# External Validation of the American College of Surgeons Surgical Risk Calculator in Elderly Patients Undergoing General Surgery Operations

**DOI:** 10.3390/jcm11237083

**Published:** 2022-11-29

**Authors:** Stamatios Kokkinakis, Alexandros Andreou, Maria Venianaki, Charito Chatzinikolaou, Emmanuel Chrysos, Konstantinos Lasithiotakis

**Affiliations:** Department of Surgery, University Hospital of Heraklion, Medical School, University of Crete, 71110 Heraklion, Greece

**Keywords:** geriatric surgery, risk assessment, validation, postoperative outcome

## Abstract

Preoperative risk stratification in the elderly surgical patient is an essential part of contemporary perioperative care and can be done with the use of the American College of Surgeons Surgical Risk Calculator (ACS-SRC). However, data on the generalizability of the ACS-SRC in the elderly is scarce. In this study, we report an external validation of the ACS-RC in a geriatric cohort. A retrospective analysis of a prospectively maintained database was performed including patients aged > 65 who underwent general surgery procedures during 2012–2017 in a Greek academic centre. The predictive ability of the ACS-SRC for post-operative outcomes was tested with the use of Brier scores, discrimination, and calibration metrics. 471 patients were included in the analysis. 30-day postoperative mortality was 3.2%. Overall, Brier scores were lower than cut-off values for almost all outcomes. Discrimination was good for serious complications (c-statistic: 0.816; 95% CI: 0.762–0.869) and death (c-statistic: 0.824; 95% CI: 0.719–0.929). The Hosmer-Lemeshow test showed good calibration for all outcomes examined. Predicted and observed length of stay (LOS) presented significant differences for emergency and for elective cases. The ACS-SRC demonstrated good predictive performance in our sample and can aid preoperative estimation of multiple outcomes except for the prediction of post-operative LOS.

## 1. Introduction

In contemporary surgical practice, geriatric patients constitute a growing proportion of the surgical population, in elective and emergency settings [1]. This patient group poses a challenge to healthcare professionals, due to the higher rate of comorbidities and influence of conditions, such as sarcopenia and frailty, which have an effect on the occurrence of post operative complications [2,3]. Routine pre-operative discussion about the perioperative risks allows for enhanced shared decision-making with the patient and their relatives, improving communication and understanding of possible outcomes, which is desirable for the elderly surgical patient [4,5]. Documentation of risk estimates can also facilitate better allocation of resources, including higher level of care and post-discharge rehabilitation services, which are essential for a large proportion of geriatric patients [6]. Risk estimation is best performed with the use of validated prediction tools.

The American College of Surgeons surgical risk calculator (ACS-SRC) is an online prediction tool, which provides estimates for multiple post-operative outcomes, based on pre-operative patient variables [7]. It was developed using clinical data from more than a million patients from multiple surgical subspecialties in the USA from 2009 to 2012. Both elective and emergency cases were included, and universal as well as procedure-specific calculators were developed. There is a lack of external validation studies for the ACS-SRC in the elderly population, with existing literature consisting of internal validations from the USA [8,9], or studies focusing on other specific surgical subgroups [10,11]. The objective of this study was to perform an external validation of the ACS-SRC for postoperative outcomes in geriatric patients undergoing general surgery.

## 2. Materials and Methods

### 2.1. Study Population

Patients 65 years or older who underwent operations within the spectrum of general surgery from 2012 to 2017 were included in this study. Procedures included elective and emergency abdominal wall hernia repairs, upper and lower gastrointestinal tract procedures, as well as soft tissue and endocrine gland operations. Written informed consent was obtained from all patients, and the study protocol was approved by the Institutional Scientific and Ethical Committee of the University Hospital of Heraklion. All patients underwent physical examination and an interview by a senior surgical trainee up to 24–48 h prior to an elective operation and immediately prior to an emergency operation. In patients with cognitive impairment, the necessary information was gathered or confirmed by their closest relative or caregiver.

### 2.2. Perioperative Data

All patients had an ACS-SRC report retrospectively completed for predicted risk assessment. Outputs of the SRC included the predicted risk of serious complications, any complication, return to the operating room, renal failure, surgical site infection (SSI), pneumonia, and death within the 30-day postoperative period. Serious complications in the ACS-SRC include cardiac arrest, myocardial infarction, pneumonia, progressive renal insufficiency, acute renal failure, pulmonary embolism, deep vein thrombosis, return to the operating room, deep incisional SSI, organ space SSI, systemic sepsis, unplanned intubation, urinary tract infection and wound disruption. Return to the operating room includes reoperation which was not planned at the time of initial surgery. Renal failure includes any rise in postoperative creatinine > 2 mg or postoperative requirement for dialysis in a patient who did not require haemodialysis preoperatively. Postoperative pneumonia diagnosis was based on radiological and clinical criteria. 

### 2.3. Statistical Analysis

Categorical variables were presented as numbers (percentage) and continuous variables are presented as mean ± standard deviation (SD) if they were normally distributed or as median with interquartile range (IQR), if they did not follow the normal distribution. Normal distribution was tested with the use of Q–Q plots and the Kolmogorov–Smirnoff test. The ACS-SRC’s predictive performance was evaluated using multiple performance metrics. C-statistic is a test that represents the area under the curve (AUC) of a receiver operating characteristic curve and it is a measure of discrimination of a prediction model [12]. Discrimination relates to how well a prediction model can discriminate those with the outcome from those without the outcome. If the SRC demonstrates perfect discriminatory performance, the AUC will be 1. If the prediction model fails to distinguish between those who will have a complication to those who will not, the AUC will be 0.5. In general, an AUC value of >0.7 indicates relatively good discrimination, and an AUC value > 0.8 indicates good discrimination. Calibration denotes the agreement between observed outcomes and predictions. For calibration, the Hosmer-Lemeshow (H-L) goodness-of-fit test was used [13]. The Brier score is a combined measure of calibration and discrimination [14]. It is reported as a score between 0 and 1. A score of 0 indicates no difference between the predicted and actual outcome, thus indicating the best possible test result. A score of 1 indicates that the test did not predict the outcome. The Brier score is compared with a Brier score cut-off, which is partially based on the incidence in the sample, meaning that a lower possible maximum value is possible for a lower incidence of the respective outcome [15]. A score above the cut-off is considered not useful. All *p*-values were two-sided, and the significance level was chosen to be 0.05. All calculations were performed with the Statistical Package for Social Sciences (SPSS) ver. 26.0 (SPSS Inc., Chicago, IL, USA).

## 3. Results

Four hundred and seventy-one patients were included in the analysis. Preoperative patient characteristics and key variables entered in the ACS-SRC are displayed in Table 1. The median age was 74 years, and the median body mass index (BMI) was 27.2. Most of the patients were American Society of Anesthesiologists (ASA) class II (48.2%), followed by ASA III (20.8%). The most common operation was cholecystectomy (28.2%) followed by lower gastrointestinal tract (GI) procedures (25.5%) and hernia repairs (22.6%). The most common operation was cholecystectomy (28.2%) followed by lower gastrointestinal tract (GI) procedures (25.5%) and hernia repairs (22.6%). Upper GI procedures included subtotal and total gastrectomies (2.5%), gastroenterostomies, and fundoplications (0.9%). Lower GI procedures mainly included colectomies (3.4% left, 5.7% right, 1.7% sigmoidectomy/Hartman’s), low anterior resections (3.4%), abdominoperineal resections (0.9%), colostomy formations (0.9%), small bowel resections (2.8%), palliative bypass (0.6%, hemorrhoidectomy (0.6%), adhesiolysis (0.6%), appendectomies (1%), and other procedures (3.6%). The rate of complications (“any complication”) reached 31.6%, 58 of them were classified as serious (12.3%), and 30-day postoperative mortality was 3.2%. Missing data rates were <2% for all variables. Discriminative performance was good for serious postoperative complications and death and relatively good for the occurrence of any complication, surgical site infection, renal failure, and pneumonia (Table 2). Brier score was 0.094 for serious postoperative complications and 0.027 for postoperative death and was lower than the calculated cut-off value for every outcome except for “any complication” as displayed in Table 2. Significant differences in the predicted and observed median length of hospital stay (LOS) were noticed, as seen in Table 3. The median observed and predicted LOS was eight days and two days, respectively (*p* < 0.001). The highest differences in the median LOS were observed in Hepatopancreatobiliary, upper, and lower gastrointestinal operations (15, 11 and 13 days respectively) and were less pronounced in emergency than in elective operations (5.5 vs. 3 days, respectively).

## 4. Discussion

In this study, we present the results of external validation of the ACS-SRC focusing on geriatric patients undergoing general surgery over five years in a Greek academic center. The predictive ability of the SRC for post-operative outcomes was assessed with the use of multiple performance measures. Overall, Brier scores, discrimination, and calibration metrics favoured the use of the ACS-SRC for the prediction of postoperative mortality and morbidity, as well as for specific postoperative complications in our geriatric cohort. The SRC’s predictive ability was low for the length of hospital stay, both in the elective and in the emergency setting.

There is a small number of studies externally validating the predictive ability of the ACS-SRC for geriatric patients, with equivocal results regarding its performance. In a recent Dutch study focusing on colorectal cancer patients, the SRC showed poor discrimination and calibration for almost all outcomes measured [10]. D’Acapito et al. performed an external validation on geriatric cholecystectomies, concluding that predictive performance was good for emergency cholecystectomies, but discrimination was low in the elective ones [11]. When multiple performance metrics were examined for elderly patients undergoing lumbar surgery, the ACS-NSQIP’s overall usefulness was low, except for moderate accuracy in predicting death [16]. The aforementioned mixed results could be partially explained by the fact that they are derived from studies focusing on specific procedures, patient samples, and health systems quite different than the one, in which the model was developed. Differences in healthcare systems may influence patient outcomes, and the Greek healthcare system is perhaps different in many aspects from its western counterparts such as lack of personnel and operative theatres, time lost for transfer to the correct level of care, absence of a cultural and legal framework for escalation of care and lack of specialized units to provide optimal care [17]. ACS-SRC is arguably the most widely used generic prognostic tool in surgery. Other novel prediction models, designed for geriatric surgical patients, have recently been mentioned in the literature [18,19]. Examples are the colorectal geriatric model (GerCRC), which used geriatric-specific predictors to estimate the risk of severe postoperative complications, and a deep neural network model from Chinese patients predicting postoperative pulmonary complications, but external validations are still lacking [18,19].

The ACS-SRC can be used in a variety of settings, both elective and urgent, for a wide range of surgical procedures and takes into consideration 21 key preoperative prognostic variables. However, a few widely accepted prognostic factors are not considered. An example is the preoperative diagnosis that prompted the surgeon to operate. The same operation can be indicated by different or multiple pathologies with significantly diverse risk profiles. Factors like hospital and surgeon volume, healthcare system, and relevant resources are also not considered. When NSQIP standards are implemented in low-middle-income countries, higher rates of adverse events highlight the need for improvement [20]. Recently, Mehaffey et al. showed that socioeconomic factors are independent predictors of postoperative outcomes and should be integrated with the ACS-SRC model [21]. All these factors might limit the generalizability of the model. 

In our study, observed and predicted length of stay differed significantly both in the entire sample and when specific procedures were analysed. The low predictive ability for LOS in our cohort can be partially explained by the lack of post-discharge facilities within the Greek healthcare system, leading to increased observed LOS, even in elective cases. The addition of geriatric consultation within the context of a standardized perioperative program has been found to result in decreased length of stay in recent reports of geriatric orthopedic and cardiac surgery patients [22,23]. Karlsson et al. reported significantly lower length of stay in Swedish elderly hip fracture patients receiving geriatric interdisciplinary home rehabilitation in the setting of a randomized controlled trial [24]. The absence of routine geriatric input and organized rehabilitation centers in Greece might imply that patients take longer to reach preoperative functional status, compared to their US counterparts in the development sample of the ACS-SRC.

Interest in improving geriatric surgical outcomes has led to organized risk stratification programmes. The ACS- National Surgery Quality Improvement Program Geriatric Surgery Pilot recently added six new variables, such as fall history and origin status on admission, specific for elderly patients, and provides risk estimates for four geriatric outcomes, such as functional decline and new mobility aid use [25]. The information provided from prediction models can be used in combination with the pre-operative patient status, to reach common ground about goals of care. Pre-operative living status, as well as transition to new caregivers post-operatively, are associated with higher rates of discharge to rehabilitation settings and increased readmissions [26]. Knowledge of these factors leads to healthcare systems, that are more prepared to deal with the complexity of the geriatric patient within organized pathways, which have been shown to improve outcomes [27].

Our study has a few limitations. It is a retrospective single-institution study, performed in a tertiary center, therefore, the sample is not representative of the entire Greek population. We included elective and emergency surgical procedures in a single-group analysis. There is a great number of factors, like limited preoperative optimization for acute-care surgery that can influence postoperative outcomes in the emergency setting. Although this methodological analysis has the drawback of increasing the heterogeneity of our sample, it is pragmatic, with results that can be generalized in an average surgical practice. Predictive performance was good when the entire cohort was considered, however, due to the small sample size, subgroup analysis based on different procedures was not possible, which could potentially identify inaccurate predictions in specific types of procedures.

The case-mix of our study was different from that, in which the ACS-NSQIP was originally developed. The satisfactory predictive performance of the model in a Southeastern European population is promising for the transportability of the calculator to different settings. Further prospective external validations of existing models should be performed focusing on the vulnerable subset of geriatric patients, and comparative validations can identify which best suits a specific healthcare system. Incorporation of risk stratification into routine geriatric perioperative practice will contribute to a higher level of care, that fits the needs of a growing surgical population.

## 5. Conclusions

The ACS-NSQIP surgical risk calculator is a valuable tool in predicting serious complications and mortality rates in elective and emergency surgical operations, in a Greek geriatric patient sample. However, its predictive ability for hospital length of stay was low. Further research to incorporate confounding factors might make results more applicable to geriatric surgery practice.

## Figures and Tables

**Table 1 jcm-11-07083-t001:** Baseline characteristics of 471 elderly patients undergoing general surgery procedures between 2012 and 2017.

Characteristic	n (%)
Age (years)	74 (10)
Female gender	209 (44.4)
Body mass index (kg/m^2^)	27.2 (6)
Preoperative functional status (missing 0.8%)	
Independent	390 (82.8)
Partially dependent	40 (8.5)
Totally dependent	37 (7.9)
ASA class (missing 2.1%)	
I	131 (27.8)
II	227 (48.2)
III	98 (20.8)
IV	4 (0.8)
Steroid use for chronic condition	6 (1.3)
Ascites within 30 days prior to surgery	0
Systemic sepsis within 48 h prior to surgery	23 (4.9)
Ventilator dependent	0
Cancer (disseminated)	25 (5.3)
Diabetes	106 (22.5)
Hypertension requiring medication	23 (4.9)
Congestive heart failure in 30 days prior to surgery	23 (4.9)
Dyspnoea with moderate exertion	23 (4.9)
Current smoker	55 (11.7)
History of severe COPD	45 (9.6)
Dialysis	3 (0.6)
Acute renal failure	12 (2.5)
Emergency case (yes/no) *	
Yes	74 (15.7)
No	393 (83.4)
Site of operation	
Hernia	102 (22.6)
Upper GI	17 (3.6)
HPB	39 (8.3)
Cholecystectomy	132 (28.2)
Lower GI	119 (25.5)
Soft tissue/thyroid/other	59 (11.8)

Data are presented as n (%) for categorical measures and as median (IQR) for continuous measures. Total number of patients in variable “Emergency case” and “Site of operation” was lower than 471 due to missing values (<2%). BMI: body mass index, ASA: American Society of Anesthesiologists, HPB: hepato-pancreato-biliary, GI: gastrointestinal, COPD: chronic obstructive pulmonary disease, OR: operating room. * according to the ACS-SRC.

**Table 2 jcm-11-07083-t002:** Performance measures of the ACS-NSQIP Risk Calculator for post-operative outcomes in Greek elderly patients undergoing general surgery procedures between 2012–2017.

Outcome	Events n (%)	Brier Score	Brier Score Cut-Off	C-Statistic (95% CI)	*p*-Value	Hosmer-Lemeshow Test
Any complication	149 (31.6%)	0.230	0.216	0.749 (0.702–0.796)	<0.001	0.063
Serious complications	58 (12.3%)	0.094	0.107	0.816 (0.762–0.869)	<0.001	0.225
Death	15 (3.2%)	0.027	0.031	0.824 (0.719–0.929)	<0.001	0.082
Return to OR	7 (1.5%)	0.015	0.015	0.639 (0.460–0.819)	<0.001	0.815
Surgical site infection	30 (6.4%)	0.056	0.059	0.763 (0.691–0.835)	<0.001	0.385
Renal failure	7 (1.5%)	0.013	0.014	0.778 (0.659–0.896)	0.019	0.297
Pneumonia	18 (3.8%)	0.037	0.037	0.789 (0.722–0.856)	<0.001	0.815

OR: Operating Room, CI: Confidence Interval.

**Table 3 jcm-11-07083-t003:** Observed versus Predicted Length of hospital stay (LOS) of 471 elderly patients who underwent general surgery procedures using the ACS-NSQIP Surgical Risk Calculator.

Procedure Type	Observed LOS (Days)	Predicted LOS (Days)	*p*-Value *
All procedures	8 (10)	2 (5)	<0.001
All emergency	9 (9)	6 (7.5)	<0.001
All elective	7 (10)	1.5 (4.5)	<0.001
Hernia	4 (4)	0.5 (0.5)	<0.001
Upper GI	18 (16)	7 (2.5)	<0.001
HPB	21 (17)	6 (3)	<0.001
Cholecystectomy	8 (10)	2 (3)	<0.001
Lower GI	13 (7)	6 (2.8)	<0.001
Soft tissue/thyroid/other	8 (10)	2 (3)	<0.001

Length of stay (LOS) is expressed as median (IQR); * Wilcoxon signed rank test; LOS: length of stay, IQR: interquartile range, HPB: hepato-pancreato-biliary, GI: gastrointestinal.

## Data Availability

All data generated or analyzed during this study are included in this published article or are available from the corresponding author upon reasonable request.

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
