# Peer review of "External Validation of the American College of Surgeons Surgical Risk Calculator in Elderly Patients Undergoing General Surgery Operations"

_jcm, 2022, doi:10.3390/jcm11237083_

Round 1
Reviewer 1 Report
Interesting research whose objective is to perform an external validation of the ACS-SRC for postoperative outcomes geriatric patients undergoing general surgery in a Greek centre.
The topic would be of interest and worth to be furthered, but I suggest that some clarifications should be made:
Lines 94-95: please explain in more detail “which is partially based on the incidence in the sample”;
Line 104: Please also define what specific surgical interventions were performed on the “Upper GI” and on the “Lower GI” tract respectively;
Line 106 and 108: please detail more precisely which complications “were classified as serious”;
Line 152 (and linked to line 171): You referred to the differences between the health systems: it would be interesting to introduce a reference to some specific elements of the health system in Greece, with a focus on surgery services;
Line 154: Perhaps it would be of interest to specify which ones would be considered “other novel prediction models”.
It is to be appreciated that the authors clearly specify the limitations of the study, and that they propose new directions for research.
Reviewer 2 Report
This is a well-written paper performing an external validation of the ACS-SRC for postoperative outcomes in geriatric patients undergoing general surgery. The authors demonstrated that the ACS-SRC showed good predictive performance and could aid preoperative estimation of multiple outcomes. I would make the following minor comments regarding the paper: 1. On the page 2 line 68-70, omit the redundant description and abbreviation of SSI. 2. On the Table 1, show the rate of emergency cases according to the ACS-SRC.
